# Life’s Mechanism

**DOI:** 10.3390/life13081750

**Published:** 2023-08-15

**Authors:** Simon Pierce

**Affiliations:** Department of Agricultural and Environmental Sciences (DiSAA), University of Milan, Via Celoria 2, 20133 Milano, Italy; simon.pierce@unimi.it; Tel.: +39-02-503-16785

**Keywords:** brownian motor, death, definition of life, feynman–smoluchowski ratchet, heat engine, theory of life

## Abstract

**Simple Summary:**

The state of ‘being alive’ is difficult to characterize because ‘life’ is currently defined using superficial features or long-term processes, rather than a single physical property unique to living things. For instance, biological molecules exhibit a vast range of structures and attributes, and a shared property is elusive. However, current knowledge suggests that key biomolecules governing a range of fundamental processes within cells do share one specific characteristic: all respond to energy absorption and dissipation by changing conformation and thus physical shape along one plane. Cyclic, repeated uniplanar shape changes induce unidirectional motion (linear or rotational movement) in molecules and the processes they govern, which is the basis of mechanistic activity and work within cells. In contrast, molecules in non-living systems do not change conformation in a way that performs work. The premise of energy conversion into directed motion suggests that life is a process whereby self-governing networks of molecular ‘heat engines’ create structure, whereas non-living structures are created and maintained by non-heat engine processes. A definition of life based on autonomous heat engine networks does not depend on any specific type of molecule or chemical process, and is potentially applicable to chemical environments different from those on Earth.

**Abstract:**

The multifarious internal workings of organisms are difficult to reconcile with a single feature defining a state of ‘being alive’. Indeed, definitions of life rely on emergent properties (growth, capacity to evolve, agency) only symptomatic of intrinsic functioning. Empirical studies demonstrate that biomolecules including ratcheting or rotating enzymes and ribozymes undergo repetitive conformation state changes driven either directly or indirectly by thermodynamic gradients. They exhibit disparate structures, but govern processes relying on directional physical motion (DNA transcription, translation, cytoskeleton transport) and share the principle of repetitive uniplanar conformation changes driven by thermodynamic gradients, producing dependable unidirectional motion: ‘heat engines’ exploiting thermodynamic disequilibria to perform work. Recognition that disparate biological molecules demonstrate conformation state changes involving directional motion, working in self-regulating networks, allows a mechanistic definition: life is a self-regulating process whereby matter undergoes cyclic, uniplanar conformation state changes that convert thermodynamic disequilibria into directed motion, performing work that locally reduces entropy. ‘Living things’ are structures including an autonomous network of units exploiting thermodynamic gradients to drive uniplanar conformation state changes that perform work. These principles are independent of any specific chemical environment, and can be applied to other biospheres.

## 1. Introduction

Life is a bewilderingly complex phenomenon involving a vast range of integrated biochemical and biophysical processes. Every cell contains millions of components performing very specific roles: biological complexity that is difficult to summarize and distil into a single defining feature. Indeed, life is typically described with a combination of properties (e.g., growth, structure, self-sustaining replication, capacity to evolve, homeostasis and metabolism) to the extent that ‘biologists now accept a laundry list of features characteristic of life rather than a unified account’ [1]. Recent thinking on the attempt to define life could even be described as defeatist [2].

More optimistically, our understanding improves with advancements in science and technology, and in light of current knowledge some of the discussion so far has proven to be relatively superficial. For example, living organisms demonstrate agency or an apparent sense of purpose (end-directed activity, also termed teleonomy), which has been suggested as the defining feature of life [3,4]. Some proponents have gone so far as to suggest that even the simplest biological organisms possess a literal, cognizant sense of purpose [5]. However, agency cannot be the distinguishing feature of life because it is not unique to biological organisms. Robot vacuum cleaners, clockwork toys and heat-seeking missiles evoke a sense of agency in human observers in precisely the same way that a turtle, a beetle or a bee would, but they do not exhibit any other features of life. This illustrates a key point: current theories and definitions fail because they focus on secondary phenomena or emergent properties without successfully discerning the underlying mechanism producing these effects.

The real enigma is whether there is a single underlying physical process from which secondary life properties emerge. As a starting point for this discussion, and as we have seen in the case of teleonomy, it is important to understand that much importance has been placed on philosophical ‘life definition problems’, but many of these are peripheral to the scientific investigation of life. These philosophical arguments are a strong voice against the endeavor, so it is important to appreciate their flaws before rolling up our sleeves and reviewing the actual data. It is also important to appreciate that a tendency to rely on longer-term, multi-generational processes to define life, such as heredity and natural selection (key theoretical frameworks in biology), can say little about the immediate state of ‘being alive’. Then, we consider empirical discoveries demonstrating a unique property of organisms that has been independently recognized in disparate contexts but has not been used to formulate a theory and definition of life, the elucidation of which is the novelty at the heart of this review.

## 2. Philosophical Barriers to Defining Life

Before setting out to discover what life is, it is important to address certain philosophical arguments that cast doubt on whether the search for an explanation of life is a realistic proposition or even a worthwhile venture [2]. A classic argument against the prospect of a scientific theory of life arises from the fact that all organisms on Earth have a common evolutionary origin. Thus, we can only observe a single type of life (*n* = 1 sample) which could even be atypical of life in the universe; we cannot know whether a theory of life truly encompasses all life-like phenomena [6]. However, scientific theories are possible explanations, supported by testable hypotheses which are accepted or rejected by observation and experiment. In other words, a scientific theory can exist so long as it has minimal empirical support, and is either refined or superseded as further hypotheses are tested. Theories have small beginnings and expand into the unknown. We have an excellent precedent that *n* = 1 is not a serious impediment to general theories of how living things operate. When Darwin and Wallace [7,8] presented their theory of evolution by means of natural selection, observational and experimental evidence was strong. Over the following decades, especially with the discovery of the structure of nucleic acids [9], with the fine details of evolutionary relationships and events revealed by genetic studies (e.g., [10,11]) and physical evidence of numerous transitional forms in the fossil record (e.g., [12,13,14]), a range of hypotheses have been tested that have increased our confidence in the theory to the point that most biologists agree that it is extremely probable (not a fact or absolute truth, per se). It provides a powerful explanation of how different types of organisms can exist, even though we can study evolution on only one planet. We are free to suggest a general biological theory based on a single biosphere, and within that biosphere can test a range of hypotheses to determine whether or not they agree with the theory.

Another contention is that definitions of life have been formulated very differently across a range of scientific disciplines, including different fields of the natural sciences and artificial life (Alife) research [2]. In fields such as astrobiology there may be various definitions for various applications, not all of which attempt to explain life. A working definition may be satisfactory for practical applications such as detecting habitable environments, whereas attempts to understand the origin of life are based on the same kind of reductive biological sciences used to scrutinize the life presently occupying the Earth, and definitions have similar theoretical goals. Definitions for Alife can only be speculative until biology has successfully explained organic life, from which to draw comparisons. This is not to say that only biology matters, rather that a realistic theory of life in organic systems would be a useful starting point for speculative considerations of life. In a sense, biology currently fails in its duty to inform other branches of science, and a lack of a clear definition of the phenomenon at the heart of biology is a major source of embarrassment. Essentially, there is good reason to attempt a theory and definition of life, and no good reason not to.

A spectrum of complexity is evident from atoms, simple chemical compounds, complex macromolecules, cells, multicellular microbes through to large-scale organisms, and the point along this spectrum at which chemistry becomes biology (abiogenesis) is difficult to identify and define, lying at the empirical and philosophical heart of the problem [15]. However, organisms, as material objects, consist of atoms and molecules and thus exhibit measurable physicochemical properties, and at every point along the spectrum scaling from atoms to organisms we now possess the methods to quantify and compare the states of matter, and have actually done so (Figure 1). Indeed, we can directly visualize in real time the movements of individual molecules [16], crucial to discerning the difference between ‘animate’ and ‘non-animate’ matter. This simple fact suggests that it is reasonable to expect that a distinguishing physical property may be detectable—a property inherent to the matter comprising organisms, yet not evident for non-biological matter—and that we can satisfy the requirement for a system and testable theory of life from which the definition of a single process emerges. In other words, we are now equipped to answer the question, “do all organisms share a single property, unique to them?”.

## 3. Long-Term vs. Immediate Life Processes

In order to address this question, it is important to recognize that some life properties occur in the longer term but others occur from moment-to-moment, and both temporal scales are often invoked in theories of life. To understand what it means for something to ‘be alive’ in any single moment it is valuable to underline why longer-term processes cannot explain this state of being alive. Indeed, heredity and natural selection (by definition, processes that require more than one generation rather than dealing with the immediate functioning of a single organism) often take center stage in the consideration of the origin, operation and definition of life [17] probably because they provide extremely strongly supported general frameworks for considering life processes. NASA’s definition states that ‘life is a self-sustaining chemical system capable of Darwinian evolution’. Similarly, a recent definition of life as ‘a self-sustaining kinetically stable dynamic reaction network derived from the replication reaction’ [15] also acknowledges the importance of longer-term events such as replication. This definition successfully consolidates many evident features of life: replication and metabolism appear to have arisen together in networks of RNA (or functionally similar) molecules catalyzing reactions for one another; life actively maintains stability by dynamic kinetic means rather than chemical inertness; molecules are variable and thus subject to natural selection, with a gradient of increasing complexity and functional effectiveness through time linking simple chemistry to the systems chemistry of living entities [15].

However, reliance on evolution and other long-term processes to define and recognize life is problematic for several reasons. We may be able to demonstrate that cells in a sample grow, multiply, produce further generations and evolve. But what if the cells are not amenable to culture? What if we cannot observe them replicating or evolving: is this because they actually are incapable of growing or evolving, or because the conditions for observation are inappropriate?

Theories of longer-term processes such as natural selection aim to explain one aspect of the natural world (in this case, how species can change through time and how divergent changes within groups can originate new species), but this is clearly a different spatial and temporal scale to the inner workings of a single organism. Indeed, life can be interpreted as an instantaneous state or short-term process, occurring moment-by-moment rather than over the timescales of generations. The fundamental importance of instantaneous processes occurring in the protoplasm (living contents) of cells has had a central role in definitions of life since the early work of Alfonso Herrera [18], who’s Plasmogenic theory states that “Life is the physicochemical activity of the protoplasm”, and that “To live is to perform a physical and chemical function. Nothing more”. These general statements would be recognized by modern biologists as essentially true, although they are not mechanistic. To understand what ‘alive’ actually means, we must be able to recognize an immediate distinguishing property characterizing the state of being alive. What is this property?

## 4. Life Reduces Entropy (But How?)

A crucial clue was provided by Erwin Schrödinger [19] when he recognized that life is characterized by the spontaneous creation of order in a universe characterized by increasing disorder, coining the term ‘negative entropy’. At its most abstract level, life is a process that orders matter in a universe in which matter and energy tend to dissipate. This is immediately evident from the fact that biological organisms use external energy and matter to accumulate organized structures (e.g., cells, tissues, bodies); locally ordered matter that literally embodies reduced entropy. Schrödinger also suggested that instructions controlling this process may be encoded in ‘aperiodic crystals’ or molecular matrices with irregular repetition of atoms encoding information, and that in some way this process may involve the chromosomes. Although our detailed knowledge has improved (DNA is a flexible polymer, not a rigid crystal) Schrödinger’s view fundamentally suggests that life is a process by which energy is used to aggregate, rearrange and organize matter, following information encoded in aperiodic molecules. This almost constitutes a definition of life, but lacks an explicit mechanism for the process by which matter is managed and reorganized. Also, the apparent paradox of a system that evidently reduces entropy without contravening the second law of thermodynamics (i.e., that entropy in a system always increases) is not explained. However, the involvement of entropy changes is a widely accepted feature of life (“the view that life is essentially an entropy economy driven by free energy converting processes enjoys a long and storied history” [20]), and entropy changes and thermodynamic (temperature/work) relationships are associated with characteristics of life such as self-organization and self-regulation even in simple chemical systems, suggesting that these considerations are crucial to abiogenesis [21].

It is clear that Earth’s cellular organisms perform work and self-organize using biological molecules, but that a wide range of different types of biological molecules are active in organizing and managing biological processes. However, it is not immediately evident that biomolecules share a single property underpinning their ability to aggregate and organize matter. It is evident that some fundamental properties are shared across a range of molecules, principally involving how they respond to the thermal environment and how they change conformation under excitation. Indeed, one of the most fundamental properties of matter is that it is always in motion and, crucially, this is especially so for biological systems. Atoms and molecules constantly vibrate and the extent to which they do so, by definition, determines the temperature of a system (atoms move even at absolute zero, due to the underlying fluctuations of uncertainty and thus zero-point energy [22]). Furthermore, thermal agitation (vibrational motion) can be exchanged by physical contact (conduction, or resonance transfer) or radiation (photon exchanges), and electrons can become ‘excited’ beyond their stable ground state. Excitation represents the temporary jump of an electron to a higher orbital and an increase in atomic radius, and thus essentially the size of the atom. As atoms change physical configuration, the molecules they compose necessarily change conformation, resulting in additional molecular motions which eventually relax with the emission of a photon and the decay of the excited state. All of these extremely rapid atomic and molecular-scale motions are crucial to physical and chemical processes. For instance, thermal agitation and the ‘storm’ of collisions amongst particles results in Brownian motion (the ‘random walk’ or motion of particles as observed in suspension [23]) and ultimately underpins phenomena such as diffusion.

Indeed, while thermal agitation, excitation and bombardment induce haphazard motions and conformation state changes in most molecules, some molecules exhibit motions that are constrained by their shape and the interactions between their component atoms: regions of the molecule (sub-units) are free to flex or rotate in only one plane. In other words, molecules exhibit an inherent range of possible conformations that are ‘sampled through motions with a topologically preferred directionality’, constrained by the properties of the molecule itself [24]. Thus, thermal agitation can induce directional motions in certain molecules, the character of which is inherent to the structure of these molecules, with conformational changes being reversible but occurring in only one plane (i.e., uniplanar) and inducing a unidirectional overall motion in the system. In fact, this is particularly evident for biological molecules.

The active domains of motor proteins can flex in specific directions (backwards and forwards in one plane), but not others [16,24,25,26], the spinning sub-units (c-subunit ring) of enzymes such as ATP synthase or V-ATPase spin in one plane [27,28] to generate ‘mechanical torque’ that performs work [29] (a complex mechanism driving or driven by, respectively, repeated uniplanar conformation shifts in additional α and β subunits), catalytic RNA molecules (ribozymes) shift between conformation states [30,31], the ribozyme components of ribosomes ratchet along mRNA to provide the driving force of protein synthesis [32,33], and RNA polymerase similarly ratchets along the DNA molecule during transcription [34]. Indeed, enzymes (catalytic proteins) exhibit conformational state changes, and the resulting physical motion is necessary to catalytic function as it facilitates substrate binding [35]. Even non-motor enzymes are known to essentially produce ‘directional mechanical force’ [36] or ‘convert chemical energy into mechanical force’ [37] to perform catalytic work; directional motion and power output are thought to be general properties of asymmetric proteins [38]. Thus, across a range of biological macromolecules, flexibility and asymmetry results in consistent, cyclic (repeated) uniplanar conformation state changes and directional mechanical action and molecular motion that can dependably perform work.

While the motion of molecules is typically inferred from structural relationships and computer modeling, we can now directly observe molecular movement, and are starting to achieve a highly detailed direct confirmation of these uniplanar conformation state changes. High speed atomic force microscopy has demonstrated the conformational motions of the myosin V motor protein, driving overall movement of the molecule along actin filament tracks as part of the mechanism changing the elongation of muscle fiber cells [16]. The myosin V molecule ‘walks’ hand-over-hand along the actin filament in what the authors describe as a ‘unidirectional processive movement’, generated by a combination of thermal excitation followed by the interaction of adenosine triphosphate (ATP) with head domains to temporarily fix them in position. These head domains change conformation in a very specific manner. Each domain can flex, but only in a single plane and to a very specific degree, described as a ‘rigid hinge’ motion [16]. The extent and direction of motion are not dependent on the surrounding context, such as interaction with the actin filament, but by the arrangement of atoms in the molecule and the conformation states possible for the head domain: slight deviation in bending would result in attachment to actin subunits at incorrect distances or directions, or in attachment to neighboring actin filaments, any of which would result in a disastrous lack of function, and the extent of conformational change is an inherent property of the molecule [16]. In this case conformation changes have been directly observed to be cyclic, strictly uniplanar and induce unidirectional motion in the system. The principal function of these motions is to generate mechanical force, which can be measured at the macroscopic scale as the force with which the muscle contracts (muscles pull in one direction because myosin conformation state changes are uniplanar and myosin ‘walks’ in one direction). This leaves no doubt that thermally-driven uniplanar molecular motions are used to perform macroscopic biological work [39].

Ribozymes, consisting of RNA, are structurally very different to motor proteins, but can nonetheless function in a similar way as enzyme-like catalysts [40]. Examples of natural ribozymes occur in a range of organisms throughout the tree of life (see [41] for review), and some are common to “Archaea, Bacteria and Eukarya” and “might be remnants of some protobiological RNA world that must have been retained because of the unique qualities of RNA that remain indispensable to life” [41]. In extant organisms catalytic activity is mediated by enzymes, but the universal presence of ribozymes across the domains of life suggests that they may have been crucial to catalysis for the organisms that preceded the Last Universal Common Ancestor (LUCA) of extant life [42]. Much of our knowledge of how they operate, or at least how they can potentially operate, comes from artificial manipulation or artificial ribozymes. For example, artificially designed ribozymes can perform ‘riboPCR’ (i.e., copy RNA templates in a manner similar to the polymerase chain reaction, PCR [40]). This range of metabolic and replicative activities is thought to be a prerequisite for abiogenesis [43,44]. Like motor proteins, ribozymes also perform these activities via directional motion. For example, the naturally occurring *Tetrahymena* ribozyme includes a mobile subunit (the ‘tP5abc three-helix junction’) which can reversibly shift between two extreme conformation states: ‘extended’ and ‘native’. Although it moves through a range of subtle intermediate states to achieve these endpoints the process essentially involves two principal conformation step changes, occurring rapidly over a period of 10 and 300 ms, respectively [45]. Thus, ribozyme function depends on a single property: the ability to reliably switch between conformation states. Just as the motion of motor proteins and other enzymes produces directional mechanical force, it is conceivable that ribozyme motions also generate and apply directional force during catalysis, although this has yet to be measured.

These are detailed views of specific biological molecules, but the processes that they govern are widely documented and are so fundamental to life that they form the basis of entire chapters of undergraduate biology textbooks (e.g., Chapter 17 of Campbell Biology [46]): DNA transcription, translation and cytoskeleton motor protein functions all involve linear unidirectional motion, and processes such as ATP synthesis involve unidirectional (albeit rotational) motion to perform work. During mitosis and meiosis, spindle microtubules slide around and are repositioned by motor protein ‘pushing’ and ‘force-locking’ [47], underpinning crucial events such as nuclear division, chromosome reduction and recombination. DNA’s role in genetics and heredity would not be viable if RNA polymerase were to randomly switch movement backwards and forwards along the DNA strand, creating disparate, non-functional segments of mRNA rather than linear mRNA transcripts. By analogy, a computer printer continuously switching the movement of the paper alternately backward and forward beneath the printhead would fail to produce an entire printed page. For many essential biological molecules, there can be little doubt that unidirectional motion (linear or rotational) based on uniplanar conformation state changes is a common principle. Thus, Schrödinger’s entropy reduction is achieved via uniplanar conformation state changes of molecules under thermal agitation, essentially converting random agitation into directed motion and thus work.

## 5. Life Is an Uphill Struggle—The Thermodynamics of Biological Molecular Machines

In the case of the linear molecular motors presented above, these can be considered, theoretically, as ‘Brownian ratchets’ [48] or ‘Feynman–Smoluchowski ratchets’ [49]: i.e., systems for converting stochasticity into order. Thermally agitated systems may include components that are free to move in one direction, but not backwards, effectively converting random movements into directional motion, akin to a ratchet composed of a rotating gear stopped by a spring-loaded pawl, driven by an agitated paddle wheel. At first glance this may seem to represent an impossible perpetual motion machine, whereby background thermal agitation is inevitably converted into continuous progressive movement (it was originally proposed as a thought experiment [49]). Indeed, when there is an even temperature across the mechanism the agitated pawl jumps and slips, and the gear has an equal probability of forward or backward rotation, and work is not possible. However, Richard Feynman [50] suggested that the probability of the gear moving in one particular direction increases if the pawl is at a lower energy state (less agitated) than the paddle wheel, i.e., with a net ‘energy input’ to the system or, more correctly, with a thermodynamic gradient or disequilibrium across the system (see also [48]). As this mechanism essentially relies on a temperature differential to perform work, Feynman et al. [50] referred to it simply as a ‘heat engine’. We know that this is possible: as a proof of principle, a physical ratcheting mechanism has been constructed that converts inputs of non-directional fluctuating forces such as white noise into unidirectional rotation (i.e., a device that spins in a noisy environment [51]).

Heat engines include any structure that uses a temperature differential between two thermal reservoirs to produce work, and the Carnot cycle [52] is a theorem describing the potential efficiency with which this can occur. Thus a ‘Carnot engine’ is an idealized, maximally efficient heat engine, whereas real heat engines are not maximally efficient. In practice, artificial, mechanical heat engines use either differentials within volumes of liquids or gases or exploit phase-changes (e.g., from liquid to gas): for example, liquid water vaporizing to increase pressure in the cylinder of a steam engine, or liquid vaporizing at increased volume/lower pressure to reduce temperature in a refrigerator. In practice, this is achieved by mechanisms that direct a bulk flow of matter between components which change the energy state: e.g., the pipes, pump, condenser and particularly the evaporator components of a refrigerator, or the boiler tubes, steam dome, steam pipe and cylinders of a reciprocating steam locomotive. Natural heat engines such as atmospheric cyclones do not have solid mechanical components, but heat flow and work (motion) similarly depend on the temperature/volume/pressure relationships of bulk flows of matter, in conceptual agreement with Carnot’s theorem. Another natural heat engine process is the formation of snowflakes, which involves a phase-change driven by a temperature gradient between the atmosphere and a nucleating body. This similarly involves the physical transfer of matter from one thermal reservoir to another, in the form of atmospheric water molecules that diffuse to and crystalize on the surface of the nucleating body [53]. For ratcheting, nanoscopic heat engines, however, the driving thermodynamic gradient does not always involve diffusion and flows of atoms: the thermodynamic gradient can occur because the atoms comprising the molecule vibrate to different degrees and a difference in agitation state across the structure sets up a thermal differential [50]. This is in general agreement with Carnot’s theorem because transfer of energy states occurs between thermal reservoirs. In other words, nanoscopic ratcheting heat engines do not operate via bulk fluxes or diffusion of atoms, but by thermodynamic gradients formed directly across their molecular structures.

Despite reducing entropy locally, nanoscopic ratcheting heat engines do not contravene the second law of thermodynamics (that entropy in a system always increases), because the work they perform represents a relatively small decrease in entropy (uphill) connected to and driven by a larger entropy increase (downhill): i.e., a localized decrease but a net increase. The driving disequilibrium across the mechanism can be thought of as an ‘environmental’ (positive entropy) disequilibrium, but the work is essentially used to create a further, weak disequilibrium (negative entropy). In simple analogy, a torrent flowing across a waterwheel (with a simple pawl to stop retrograde motion) operates a pulley system to lift a bucket of water uphill: a small mass of water can move against gravitational attraction to the Earth because it is driven by a much larger mass that moves with gravity. More precisely, these irreversible heat engine mechanisms are akin to the escapement of a clock, in which the kinetic energy of a rotating gear is alternately restrained by, then pushes, an oscillating pendulum [54]. A simple force is regulated to produce a precise movement, and the entire mechanism can only work with the simultaneous interleaving of both input and output actions [54,55]. Another useful analogy is that of a two-way turnstile, in which action is regulated both by a major, driving disequilibrium and a weaker, driven disequilibrium (a ‘free energy conversion (FEC) turnstile coupling device’; [20]). The ‘downhill’ (toward thermodynamic equilibrium) gradient is both regulated by and drives the ‘uphill’ (entropy reducing) gradient. Living systems are uphill systems, but can only exist in a downhill environment, necessarily exploiting thermodynamic gradients and a net entropy increase [54].

Here a distinction should be made between the thermodynamics of molecular motors (i.e., ratcheting, irreversible heat engines exploiting thermodynamic gradients across their structure) and of reversibly rotating enzymes such as ATP synthase which, being driven by electrochemical gradients, are not generally considered to be heat engines per se. The driving force is not a thermodynamic gradient operating across the structure of the molecule itself, but the trans-membrane electrochemical gradient of protons in solution (the proton-motive force operating during the process of chemiosmosis). However, this is similar to the type of classical heat engines that exploit differences in a single phase of matter and bulk flow or diffusion between thermal reservoirs. The driving forces underpinning the proton-motive force are diffusion (along the chemical gradient) and the electrostatic force (along the electric gradient). These are thermodynamic processes—the motive force of diffusion is ultimately (from an atomistic point of view) the random walk of particles propelled by the bombardment of thermally agitated particles of the medium (i.e., Brownian motion). Motion tends to occur towards zones of lower solute concentration because there is a lower probability of occupied space and greater freedom of movement. For rotary enzymes, the two thermal reservoirs are the compartments on either side of the membrane, and they can be considered ‘Brownian diffusion machines’ that exploit a thermodynamic gradient and thus ultimately thermal agitation. They thus constitute a type of heat engine, although one lacking an inherent ratcheting mechanism and indirectly exploiting a complex thermodynamic gradient also mediated by the electrostatic force. In the case of ATP synthase, this is likely to have been a key adaptation exhibited by the Last Universal Common Ancestor, evolving from enzymes that transported proteins and, originally, RNA across the membrane [56]. Indeed, life preceding the LUCA was probably based on the ability to exploit proton gradients [57] which is widely seen as a trait central to abiogenesis [58]. Although chemiosmosis is usually considered in terms of electrochemistry, it is important to acknowledge the underlying role of thermodynamics in providing the motive force. Crucially, rotary enzymes and ratcheting biomolecules share the fundamental principle of exploiting nanoscale thermodynamic gradients to drive uniplanar conformation state changes, favoring reactions that have directionality and can thus perform work. Some, such as V-ATPase, perform the opposite function of using ATP-induced uniplanar conformation state changes ultimately to create electrochemical gradients, but the ATP used is a temporary carrier of energy stored from the prior exploitation of an initial driving thermodynamic gradient.

What does ‘energy carrier’ or ‘chemical energy’ actually mean in the context of molecules such as ATP? Crucially, while thermal agitation is the torrent that induces motion [59], ATP acts essentially by fixing the motion of biomolecules at a point far from thermodynamic equilibrium (i.e., ATP carries a disequilibrium [54,60,61]). In other words, molecules such as ATP are missing components of biological heat engines, required to temporarily complete the configuration, induce the disequilibrium across the structure and thereby activate it, with the subsequent motion and work then resetting the configuration.

Many of these concepts have previously been acknowledged as fundamental to life [34,54,62], and the central role of thermodynamic disequilibria utilization in particular as an essential and distinguishing property of life has already been recognized, forming the basis of the ‘alkaline hydrothermal model’ for the emergence of life on Earth (hydrothermal serpentine mounds may have provided the thermodynamic gradients, compartments, reactants and, crucially, interleaved specific organized structures required by proto-biology [20]). Indeed, Branscomb and Russell [20] discuss a hypothetical turnstile coupling device involved in the origin of metabolism, suggesting a proton pump which may have involved components that “rotationally flex” to move protons across a membranous interface against an electrochemical gradient. While these concepts are thus well established, the novelty of the present discussion rests in the fact that the principle of uniplanar conformation state changes directing thermal agitation as the driving mechanism reducing local entropy has not been used to formulate an explicit theory or definition of life.

## 6. The Single Property Defining Living Systems

The structurally diverse biological macromolecules discussed above exhibit a shared principle of operation: that of conformation state changes directing thermodynamic disequilibria into unidirectional motion and thus work (local entropy reduction). Alternatively, molecules without preferred configuration state changes move randomly, dissipate energy inputs and are not involved in performing work. This simple functional difference suggests the existence of two fundamental functional classes of matter (‘energy directing’ or ‘energy dissipating’; Figure 2), forming the basis of the difference between living and non-living systems. Life can be defined thus:


*Life is a self-regulating process whereby matter undergoes cyclic, uniplanar conformation state changes that convert thermodynamic disequilibria into directed motion, performing work that locally reduces entropy.*


**Figure 2 life-13-01750-f002:**
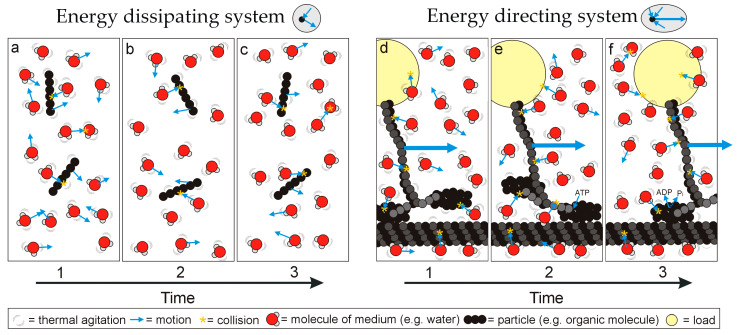
Simplified representation of how different types of matter respond to the chaotic thermal environment: matter either exhibits (**a**–**c**) a banal conformation that moves randomly under thermal agitation and bombardment, dissipating energy, or (**d**–**f**) uniplanar conformation changes that convert thermal agitation or excitation into directed motion, as part of an energy directing system that can perform work over time (the representation is inspired by a motor protein ‘walking’ along a microtubule to pull a load). Note that the motive force is Brownian motion (i.e., thermal agitation of the particles themselves but also bombardment by molecules of the surrounding medium; heat engine function must be considered in the context of the medium). For the energy directing system, ATP has an additional role in completing the conformation, affecting the thermodynamic disequilibrium across the particle and preventing backwards motion.

Self-regulation via integrated networks [15] and autonomy [63] are key concepts highlighted in this definition. It is not the single protein (the single heat engine) that should be considered alive, but the integrated, self-regulating and self-replicating network of heat engines. For example, looms use cyclic conformation changes (mechanical action) to convert energy and matter (electricity and wool) into an ordered state (cloth) following a pattern encoded as a set of instructions (programmed information). However, looms are not self-regulating systems and require external input (from a biological organism) for their creation, maintenance, operation and programming. Thus, if we wish to classify an object as alive, the definition of a living thing is: a structure comprising, at least in part, an autonomous network of units exploiting thermodynamic gradients to drive uniplanar conformation state changes that perform work.

Indeed, self-regulation also encompasses the process of self-replication. Mules, dogs, humans, plants, bacteria, archaea all rely on networks of heat engines performing work and replicating within them. Organisms are ‘alive’ from one moment to the next due to the operation of heat engines. Within each of your cells, millions of heat engines continuously jiggle, bathed in thermal energy and continually bombarded by water molecules, performing small tasks so numerous and rapid that the sum allows the operation of metabolism, physiology, movement, growth, reproduction, and all the emergent characteristics that we traditionally use to define life. As living beings, this is our defining physical interaction with the universe; the single distinctive property distinguishing ‘living’ from ‘non-living’ things.

The process encompassed by the definition determines the immediate state of being alive, agrees with the concept of disequilibria driving Feynman–Smoluchowski Brownian ratchets [48,54], is a mechanism that aggregates matter to produce negative entropy [19], underpins the ‘self-sustaining kinetically stable dynamic reaction network derived from the replication reaction’ [15], its components are subject to the further long-term processes of mutation and natural selection [7,8], and it agrees with the ‘plasmogenic’ view of life as the physicochemical reactions occurring in the protoplasm [18]: the definition is thus consistent with a range of fundamental biological and physical concepts. Lack of coordinated, directed motion in matter reflects a state of non-life, and where directed motion was previously evident in a molecular network, this lack essentially determines death. ‘Animate matter’ really is an appropriate lay description for the essential process underpinning life, albeit one that does not quite capture the range of scales (nanoscopic to macroscopic) and the intricacy of the processes involved.

## 7. Falsification and Rejection

Rejection of the above theory and definition of life hinges on a rigorous and convincing falsification, such as an unambiguous exception to the rule [2]. Simple mechanisms, such as the device that spins in a noisy environment [51] are not involved in networks that create structure and reduce entropy, and do not satisfy the definition (they are not alive). Traditional exceptions to life definitions, such as fire, cyclones and crystals, do not involve entropy reduction by heat engines (they are not exceptions; they are not alive). Fire is a self-sustaining reaction but increases entropy. Cyclones show structure and, as discussed above, are themselves single heat engines, but structure emerges from convection and pressure gradients rather than uniplanar conformation state changes within the matter from which they are composed, and they are not involved in maintaining a stable autonomous network. Diamonds, table salt and snowflakes exhibit growth, structure and entropy decrease during formation, but crystallization results from compaction at high temperature, precipitation from a solution, or by freezing of vapour, respectively, rather than being products of an autonomous and integrated network of heat engines.

Bacteria frozen in the permafrost or tardigrades frozen on Antarctic moss are alive, because metabolism (working on heat engine principles) does proceed, albeit extremely slowly, with cell components in a protected state known as cryptobiosis [64]. Cryptobiosis, in which high concentrations of sugars and heat shock proteins are mobilized to physically support and thus protect biological molecules (including structures such as cell membranes, enzymes and DNA) is a widespread and well-studied phenomenon [65]. For example, plant embryos remain inactive but viable within seeds due to the ‘chaperone’ properties of proteins such as late embryogenesis abundant (LEA) proteins, heat- and cold- shock proteins and sugars; part of a universal cellular stress response that is evident to differing degrees in all organisms [66]. Most cells are capable of a degree of inactivity, crucial to survival of stress (i.e., sub-optimal metabolic performance imposed by variable or limiting environmental conditions [67]).

Red blood cells (erythrocytes) require an active metabolism in order to maintain the integrity and function of the cell membrane and of the hemoglobin that holds the oxygen they transport. The cytoskeleton (with its associated ratcheting motor enzymes) is an essential component working to stabilize the membrane, but also maintain the correct flexibility. In the context of the above definition of life, erythrocytes function and live in an instantaneous sense, and die when the internal network of molecular motors ceases to function. Mammal erythrocytes do not include a nucleus or organelles, lacking some cell functions such as protein synthesis and oxidative phosphorylation, thereby limiting their autonomy and ability to persist. This has several advantages for mammal erythrocytes, including the ability to efficiently change shape as they pass through capillaries and, lacking the machinery required for replication, the superpower of invulnerability to viral infection. Aside from mammals and a few amphibians the erythrocytes of most animal groups do exhibit a nucleus and organelles. Bird erythrocytes, for example, have working mitochondria [68] and fish erythrocytes are known to perform protein synthesis [69], although they do not replicate autonomously and are produced in an organ equivalent to the kidney (the opisthonephros). While it is undoubtedly correct to refer to the precursor cells of erythrocytes (normoblasts) as alive, mature erythrocytes should perhaps be seen as senescent (i.e., alive but no longer capable of a full suite of synthesis and replicative functions, and thus persistence). The same reasoning could be applied to other non-replicating cell types such as neurons. For example, a nervous system is alive but neuron function precludes mitosis and cellular replication, so the nervous system is inherently senescent; replication of the entire organism is required to generate a fresh nervous system. Organisms that lack nervous systems, such as plants, do not have the limitations (or advantages) of neurons, and can grow indeterminately.

Prions (misfolded prion protein; PrPSc) have biological origins and appear to replicate, but are structurally rigid (the conformation changes occurring during their formation are akin to an irreversible collapse and crumpling [70]), and the ‘replication’ induced by PrPSc has little to do with true replication (i.e., production of new complex structures from simpler materials following information inherited across generations). PrPSc does not create, but alters the state of existing protein. Specifically, ‘cellular prion protein’ (PrPC; a nerve cell membrane transporter protein [71]) is altered in a way that happens to induce a cascade of further damage and conversion of PrPC to PrPSc. Furthermore, PrPSc does not participate in a network that locally reduces entropy to create structure, but leads to tissue destruction and increasingly disordered states, increasing entropy. In other words, if prions are considered in the context of the above definition, they do not falsify it. They are not a ‘biological exception’ to the rule. They are simply not alive.

Neither do viruses represent an exception, but truly bridge the gap between life and non-life, because in their free state they are aggregates of molecules (a non-living state), but when they encounter cell membranes and are then intimately incorporated into metabolic machinery, they actively participate in the directed motion network (share the living state of the cell), which reduces entropy by converting simple resources into more complex copies of virus particles. Life is a process that can stop and start. Abiogenesis—chemistry becoming biology—should not be considered a single mystic event that happened just once billions of years ago; viruses perform their version of this trick every day.

Medical definitions of life and death are particularly interesting in the context of the above definition, because they are directly compatible with it, although representing states and consequences occurring at the macroscopic scale, immediately evident to a qualified human observer. In the USA, the Uniform Determination of Death Act (UDDA) states that an individual who has sustained either (1) irreversible cessation of circulatory and respiratory functions, or (2) irreversible cessation of all functions of the entire brain, including the brain stem, is dead. These are practical criteria that are intended to allow a legal definition of death. However, they reflect underlying biological processes, death being the moment when integration of heat engine networks ceases in (1) the heart or (2) the brain. Human bodies are a mosaic of life and non-life, meaning that medical death of the person (the entire organism) can be ascribed based on the irreversible failure of one vital organ (heart or brain) despite other organs being alive. In the case of live organ transplants, a living heart (with cells demonstrating active and integrated heat engines) removed from a donor with a dead brain (in which heat engine integration is quenched) is congruent with the definition of life, the medical state simply representing the underlying biological/physical state.

Can artificial systems or constructs falsify the above definition? Brownian ratchets, or conceptual equivalents, are found in artificial systems such as liquid crystal displays [72], diodes (which impart unidirectionality on electrical current) or devices such as electronic switches that sort suspended particles [73], and a range of artificial nanoscale Brownian motion devices have been constructed [62]. However, by definition artificial systems do not build themselves. If an artificial network of devices were able to use a heat engine network to reduce entropy, create order and subsequently become self-regulating and self-replicating, then it would not falsify the definition; it would then be considered alive.

## 8. Other Potential Forms of Life

Of the various forms of artificial life, based on hardware, software or artificial cells (‘hard’, ‘soft’ and ‘wet’ Alife, respectively [74]), digital software organisms seem the most far-removed from a definition of life based on matter. However, even computer software has a physical basis in the states (the presence or absence of charge and thus bits) of memory cells and the distribution of these states (physical addresses) across a memory chip. Complications exist, such as when states are represented indirectly in ‘virtual memory’ (distributed on the hard disc rather than arrayed on the memory chip), but the term entropy is used to represent the extent to which processes are physically distributed across hardware [75]. A virtual environment modeling unstructured systems such as a dust cloud will not only represent a high-entropy system, it will also literally exhibit higher entropy in the state of the memory chip in the real world. In comparison, a highly ordered virtual reality would exhibit relatively low entropy even in the real world, as a structured distribution of memory cell states. Software code induces physical state changes in material hardware, and digital structures have a direct foundation in the material world. Software has a physical entropy state.

Constructs in virtual space (polygon meshes) are physically stored as arrays of bits on the memory chip, but are conceptually similar to molecules in that they are essentially geometric forms exhibiting properties of flexibility, restriction of movement and interaction with other forms (dynamic geometry). If a simulated network of ‘dynamic geometry molecules’ were to operate in a way that exploited a simulated non-equilibrium state such as a ‘heat’ difference (difference in agitation states) to induce unidirectional motion and create ordered states, then it would reduce entropy in both virtual and real space and operate in essentially the same way as a biological organism. While detailed modeling of single heat engines is currently possible [76,77,78], simulation of complex networks of units with roles in replication and metabolism would be a greater technical challenge in terms of processing power. Eventually, one can conceive of a ‘soft’ ALife system managing and feeding back with a ‘hard’ ALife system to create a self-sustaining and self-governing physical structure. This is conceptually similar to the mechanics of a large multicellular organism functioning under the influence of biochemistry and instructions operating at much smaller physical scales. Indeed, many biological organisms are composed of structures operating on different principles over vastly different scales, from molecules, cells, tissues, to organs, integrated to allow self-sufficiency and survival of the individual. Populations of Alife systems could also be subject to ‘virtual selection’, as errors in virtual nucleic acid sequences could create virtual mutations, affecting the construction of hardware, with only the fittest (most appropriately functioning) survivors able to construct further copies.

Thus, the biological definition of life suggested above may at first seem far removed from the field of Alife, but may find increasing relevance if artificial networks of soft and hard components using the heat engine principle can organize resources and become self-reliant, directly analogous to organisms. If this actually transpires, a key philosophical dilemma will be whether this can be considered ‘artificial’ or not, or whether a self-replicating phenomenon represents a post-artificial case of *n* = 2. Other dilemmas may include epidemiological considerations and quarantine measures.

## 9. Conclusions

Life represents order emerging from molecular uniplanar conformation state changes that direct thermal agitation and excitation energy into catalysis of reactions perpetuating a negative entropy replication network. Life’s main requirement is the thermal bath and increasing entropy of the universe, and thermal agitation is particularly strong in the regions of the universe close to stars. Many star systems are now known to include planets with an appropriate temperature such that liquid water and complex molecules almost certainly exist [79,80]. As the difference between living and non-living matter rests in differences in configuration under thermodynamic agitation, simple life forms—identifiable as such because their components change conformation states cyclically to perform tasks together in self-replicating networks—are likely to be extremely common throughout the universe. If a sample from another planetary body demonstrates organized structure associated with a suite of components operating on the heat engine principle within a thermally agitated medium, it would be a strong indicator of life.

## Figures and Tables

**Figure 1 life-13-01750-f001:**
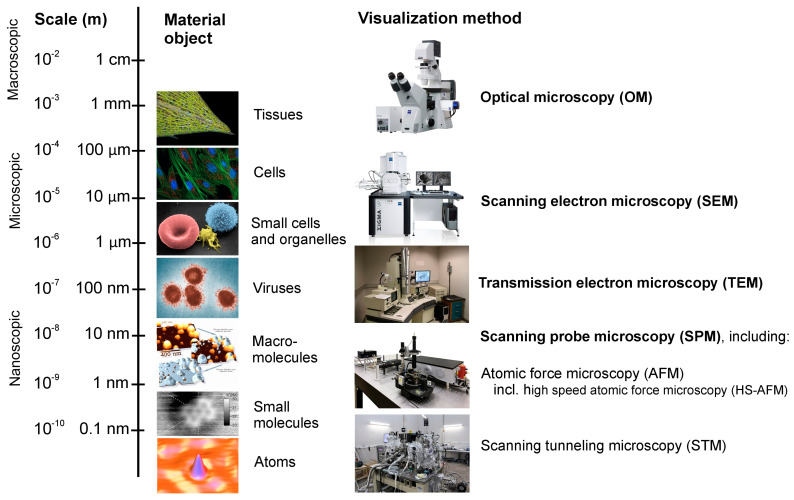
The properties of matter can now be investigated and visualized across a wide range of spatial scales, from macroscopic objects to atoms, promising a mechanistic explanation of living things as material objects. Individual images are credited in descending order by (1). material object (Karl Gaff; ZEISS Microscopy; Electron Microscopy Facility at The National Cancer Institute at Frederick; Guest2625; Y. Roiter, M. Ornatska, A. R. Rammohan, J. Balakrishnan, D. R. Heine, and S. Minko; Kota Iwata; NIST, Joseph Stroscio) and (2). visualization method (Zeiss Microscopy; Zeiss; Oak tree road; AAMonitor96; Rickinasia) and are public domain or published under a Creative Commons Attribution-Share Alike 2.0, 3.0 or 4.0 license at commons.wikimedia.org, accessed on 8 July 2023).

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
