# Peer review of "Life’s Mechanism"

_life, 2023, doi:10.3390/life13081750_

Round 1

Reviewer 1 Report

The ms by Simon Pierce proposes a more specific version of Astumian’s, Branscomb’s and others’ scenario that life’s characterizing feature is a Brownian-ratchet-type heat-engine mechanism. The ms builds upon this specific version to propose a unifying definition of life.

It is refreshing to see such kind of work taking the thermodynamic hurdles in explaining life seriously, while the vast majority of articles in the origin-of-life field fantasizes around some magical spontaneous transition from “primordial” organic molecules towards the extremely low entropy state of a living cell – a thermodynamic non-sense!

I therefore tend to advocate publication of this ms. However, I have doubts about the actual reality of several claims made which the author clearly needs to address before I can fully endorse this ms. Furthermore, the ms would benefit from some streamlining as I will also point out below (I do feel that this ms is unfortunately written in a somewhat chaotic manner).

- The main claim: the uniplanar conformational changes

In paragraph 4 (lines 214-350) the author presents biological examples (enzymes) that underlie his general hypothesis. These are chlorophyll-based systems, ATP synthases, ribozymes, RNA polymerases and actin/myosin motors. I admit being pretty ignorant when it comes to RNA polymerases and the muscle motors. I would contend that ribozymes (lines 318-332) are irrelevant to the topic of this ms since they are artificial products (I don’t know of any “genuine” ribozyme in extant life).

By contrast, I think that I know a bit about photosynthesis and ATP synthase and what I know is in blatant conflict to the claims of this ms! I would go as far as saying that lines 256-266 will make any photosynthesis researcher’s hair stand on end. Excitation transfer is energetically downhill through a fixed scaffold of pigments, directionality is only imposed by the absorption/emission wavelengths of individual pigments which is set by the protein environment. Nothing is moving here (apart from the vibrational motion inherent to all matter above 0 K). The author cites an obscure paper to substantiate his claim. I dug out this work and tried to make sense of it. It clearly is one of the most cryptically written articles I have come across, but even with my best intentions I cannot find any mentioning of the specific movement of Chl the author describes in his ms. I can only guess that he may have been mistaken by the last (schematic) figure in this article, where two Chls are shown within the framework of the LHC-protein. These are NOT two distinct conformations of a pigment but represent individual Chla and Chlb moieties. I hold that there isn’t anything “swiveling” neither in excitation transfer nor in charge separation and transfer within the photosynthetic reaction centres! By the way, the remainder of the author’s description of the photosynthetic reaction chain (lines 259-266), although not pertinent to the topic of this ms, is equally misguided. In summary, the author’s statement that “this mechanism of the conversion of electromagnetic energy to chemical energy … is reliant on molecules physically changing shape in a specific and repetitive manner …” is not supported by any experimental evidence.

- ATP synthase

The author doesn’t go into details of ATP synthase’s functional principle but only says that it’s textbook knowledge. Ok, but what does this knowledge say? It says that there are actually two distinct places on the entire enzyme where conformational movements occur, one in the membrane-integral ring as compared to the stalk and another one on the ATP/ADP-binding subunits of the coupling factor complex. Both these movements aren't Brownian ratchets but their directionality is determined for the former by the sign and magnitude of the transmembrane proton-motive-force and for the latter by the ATP/ADP ratio. ATP synthases are fully reversible and “ratchets” are only induced by chemical modification (mainly disulfide bridge chemistry) which are long-time-range modifications (e.g. during the night in photosynthetic organisms or organelles or in root organelles). I therefore don’t see how they can serve to corroborate the author’s hypothesis.

Minor problems:

- The ms is verbose and large parts such as pages 2 to 4 are superfluous at best and contentious in many places. For example, the entire argument on lines 64-78 are very debatable. Your argument that reproduction cannot be used in definitions of what it is to be alive is to my mind misguided. Vasectomized men as well as mules are alive because their cells permanently reproduce. Once your cells stop renewing themselves, your days are severely counted.

I also don’t see what all the discussions on the “dog-problem” have to do with the topic of this ms.

- The author’s musings on lines 191-200 are misconceptions. We didn’t know there were Archaea out there not because they couldn't be cultured but because we didn’t have the discriminating criterion, i.e. the phylogenetic divergence of their 16S rRNA sequences. Many archaeal species were known long before but could not be recognized as such as long as we didn’t have these sequence information.

- I am afraid that you affirmation that red blood cells are “alive” would meet with many very pertinent counter-arguments if exposed to the origin-of-life community ;-)

Author Response

Reviewer 1

The ms by Simon Pierce proposes a more specific version of Astumian’s, Branscomb’s and others’ scenario that life’s characterizing feature is a Brownian-ratchet-type heat-engine mechanism. The ms builds upon this specific version to propose a unifying definition of life.

It is refreshing to see such kind of work taking the thermodynamic hurdles in explaining life seriously, while the vast majority of articles in the origin-of-life field fantasizes around some magical spontaneous transition from “primordial” organic molecules towards the extremely low entropy state of a living cell – a thermodynamic non-sense!

I therefore tend to advocate publication of this ms. However, I have doubts about the actual reality of several claims made which the author clearly needs to address before I can fully endorse this ms. Furthermore, the ms would benefit from some streamlining as I will also point out below (I do feel that this ms is unfortunately written in a somewhat chaotic manner).

- The main claim: the uniplanar conformational changes

In paragraph 4 (lines 214-350) the author presents biological examples (enzymes) that underlie his general hypothesis. These are chlorophyll-based systems, ATP synthases, ribozymes, RNA polymerases and actin/myosin motors. I admit being pretty ignorant when it comes to RNA polymerases and the muscle motors. I would contend that ribozymes (lines 318-332) are irrelevant to the topic of this ms since they are artificial products (I don’t know of any “genuine” ribozyme in extant life).

AUTHOR: The example I used is a naturally occurring ribozyme from the protist species Tetrahymena thermophila. A review of natural ribozymes in a range of organisms including cherry trees, carnation plants, Dolichopoda cave crickets, tobacco ringspot virus, chicory yellow mottle virus, arabis mosaic virus, Neurospora spp., gram-positive bacteria and mitochondria is now included (Talin et al. 2009; https://doi.org/10.1016/j.resmic.2009.05.005 ), including the observation that some are common to “Archaea, Bacteria and Eukarya”. (Similar themes are explored in this Nature paper: https://www-nature-com.pros1.lib.unimi.it/articles/418222a.) Aside from natural ribozymes, there is a huge literature on artificial ribozymes and the fact that they can be coaxed into more complex functions than are seen in natural ribozymes, and these should be seen as (and are now explicitly presented as) examples of what ribozymes could be capable of, and that it is reasonable to expect they could have had these roles in an ancient RNA world. Crucially, the following statement remains true: natural ribozymes exist and ribozyme catalysis is known to involve directional conformation state changes. I have revised the text to include an additional sentence to include these citations to reviews of natural ribozymes.

By contrast, I think that I know a bit about photosynthesis and ATP synthase and what I know is in blatant conflict to the claims of this ms! I would go as far as saying that lines 256-266 will make any photosynthesis researcher’s hair stand on end. Excitation transfer is energetically downhill through a fixed scaffold of pigments, directionality is only imposed by the absorption/emission wavelengths of individual pigments which is set by the protein environment. Nothing is moving here (apart from the vibrational motion inherent to all matter above 0 K). The author cites an obscure paper to substantiate his claim. I dug out this work and tried to make sense of it. It clearly is one of the most cryptically written articles I have come across, but even with my best intentions I cannot find any mentioning of the specific movement of Chl the author describes in his ms. I can only guess that he may have been mistaken by the last (schematic) figure in this article, where two Chls are shown within the framework of the LHC-protein. These are NOT two distinct conformations of a pigment but represent individual Chla and Chlb moieties. I hold that there isn’t anything “swiveling” neither in excitation transfer nor in charge separation and transfer within the photosynthetic reaction centres! By the way, the remainder of the author’s description of the photosynthetic reaction chain (lines 259-266), although not pertinent to the topic of this ms, is equally misguided. In summary, the author’s statement that “this mechanism of the conversion of electromagnetic energy to chemical energy … is reliant on molecules physically changing shape in a specific and repetitive manner …” is not supported by any experimental evidence.

AUTHOR: The reviewer is correct, the example has been eliminated, and I can explain why I made this mistake. Firstly, the cited paper was published in a peer-reviewed journal with an impact factor >2, and I had no reason not to trust this. The main object of study (as stated in the title) was the ‘internal motion of chlorophyll a’: “In this study, the internal motion of chlorophyll a of LHCII in the lipid bilayer membrane was investigated”. It documents ‘motional components’, ‘orientations’ and ‘segmented rotation’ of chlorophyll. However, it was indeed a “cryptically written” paper which I apparently over-interpreted. Also, the authors used a simplified membrane system, not the thylakoid membrane, and the only link to in vivo processes seems to be that they were inspired by the natural system (but actually used a simplified artificial system). Like the reviewer, I too cannot find anything that specifically states if or how this occurs in vivo or if it is related to resonance energy transfer, which was a claim I had made in my manuscript. So, I searched for other papers on this subject and none mention rotational motion of chlorophyll during resonance energy transfer, and indeed they show that because the chlorophylls are sufficiently close together the energy state is transferred directly across them as a delocalized wave packet or exciton, with no need for the kind of ‘segmented rotation’ movements that seemed to be suggested (e.g. https://doi.org/10.1016/j.tplants.2018.03.007). Chlorophyll does vibrate during excitation and relaxation (https://doi.org/10.3390/ijms21082836v), but it is the positioning of the chlorophylls (they are lined up) that allows the exciton to transfer across them to the reaction center (https://pubs.acs.org/doi/pdf/10.1021/jp960486z) and I find no evidence of swiveling or bending motions of the molecules themselves. Essentially, I had used a single citation that is not in agreement with other studies, from which I made a strong claim - it is therefore correct that it has been eliminated from the revised manuscript.

- ATP synthase

The author doesn’t go into details of ATP synthase’s functional principle but only says that it’s textbook knowledge. Ok, but what does this knowledge say? It says that there are actually two distinct places on the entire enzyme where conformational movements occur, one in the membrane-integral ring as compared to the stalk and another one on the ATP/ADP-binding subunits of the coupling factor complex. Both these movements aren't Brownian ratchets but their directionality is determined for the former by the sign and magnitude of the transmembrane proton-motive-force and for the latter by the ATP/ADP ratio. ATP synthases are fully reversible and “ratchets” are only induced by chemical modification (mainly disulfide bridge chemistry) which are long-time-range modifications (e.g. during the night in photosynthetic organisms or organelles or in root organelles). I therefore don’t see how they can serve to corroborate the author’s hypothesis.

AUTHOR: My error was to state (at the start of the section on thermodynamics) that “The real biological molecules presented above can all be considered, theoretically, as ‘Brownian ratchets’”. This statement is incorrect because, as the reviewer points out, ATP synthase lacks a blocking mechanism or ‘pawl’ to stop reverse motion, so it is not a ratchet (i.e. ratcheting Brownian machine). ATP synthase does, however, exploit Brownian motion, albeit indirectly, and is a molecular machine that essentially converts random thermal agitation into directional motion to do work. This is best explained by the new text, the first part of which is more specific concerning the type of conformation changes seen in rotary enzymes (as suggested by the reviewer):

“the spinning sub-units (c-subunit ring) of enzymes such as ATP synthase or V-ATPase spin in one plane [26, 27] to generate ‘mechanical torque’ that performs work [28] (driving or driven by, respectively, repeated uniplanar conformation shifts in the α and β subunits),”

… and …

“Here a distinction should be made between the thermodynamics of molecular motors (i.e. ratcheting, irreversible heat engines exploiting thermodynamic gradients across their structure) and of reversibly rotating enzymes such as ATP synthase which, being driven by electrochemical gradients, are not generally considered to be heat engines per se. The driving force is not a thermodynamic gradient operating across the structure of the molecule itself, but the trans-membrane electrochemical gradient of protons in solution. However, this is similar to the type of classical heat engines that exploit differences in a single phase of matter and bulk flow or diffusion between thermal reservoirs. The driving force of electrochemical gradients is diffusion, and diffusion is a thermodynamic process – the motive force is ultimately (from an atomistic point of view) the random walk of particles propelled by the bombardment of thermally agitated atoms in the medium (i.e., Brownian motion). Motion tends to occur towards zones of lower solute concentration because there is a lower probability of occupied space and greater freedom of movement. For rotary enzymes, the two thermal reservoirs are the compartments on either side of the membrane, and they can be considered ‘Brownian diffusion machines’ that exploit a thermodynamic gradient and thus ultimately thermal agitation. They thus constitute a type of heat engine, although one lacking an inherent ratcheting mechanism and exploiting the thermodynamic gradient indirectly (they bridge the thermal reservoirs, rather than including the reservoirs in their structure as do ratcheting heat engines). In the case of ATP synthase, this is likely to have been a key adaptation exhibited by the Last Universal Common Ancestor (LUCA) of extant life, evolving from enzymes that transported proteins and, originally, RNA, across the membrane [55]. Indeed, life preceding LUCA was probably based on the ability to exploit proton gradients [56] which is widely seen as a trait central to abiogenesis [57]. Although proton gradients (and mechanisms that exploit these) are usually considered in terms of electrochemistry, it is important to acknowledge the underlying role of thermodynamics in providing the motive force. Crucially, rotary enzymes and ratcheting biomolecules share the fundamental principle of exploiting nanoscale thermodynamic gradients to drive uniplanar conformation state changes, favouring reactions that have directionality and can thus perform work. Some, such as V-ATPase, perform the opposite function of using ATP-induced uniplanar conformation state changes to create electrochemical gradients, but the ATP used is a temporary carrier of energy stored from the prior exploitation of an initial driving thermodynamic gradient. ”

Minor problems:

- The ms is verbose and large parts such as pages 2 to 4 are superfluous at best and contentious in many places. For example, the entire argument on lines 64-78 are very debatable. Your argument that reproduction cannot be used in definitions of what it is to be alive is to my mind misguided. Vasectomized men as well as mules are alive because their cells permanently reproduce. Once your cells stop renewing themselves, your days are severely counted.

AUTHOR:  I agree and have removed these complicating and contentious statements.

I also don’t see what all the discussions on the “dog-problem” have to do with the topic of this ms.

AUTHOR: I have removed these two paragraphs, and this part of the manuscript now flows much better. My original aim had been to address classic conundrums that are very often promoted by researchers with a strictly philosophical approach to the life definition problem (essentially, you cannot define ‘dog’, so definition of ‘life’ is not possible). At the end of the day these are also arguments without any consideration of biology, including the fact that biologists can (as was detailed) precisely define ‘dog’. I felt I had to deal with them, but I don’t necessarily have to. It is refreshing to simply ignore them as being of limited relevance, and I am happy to eliminate them.

- The author’s musings on lines 191-200 are misconceptions. We didn’t know there were Archaea out there not because they couldn't be cultured but because we didn’t have the discriminating criterion, i.e. the phylogenetic divergence of their 16S rRNA sequences. Many archaeal species were known long before but could not be recognized as such as long as we didn’t have these sequence information.

AUTHOR: Good point. I have removed this example. However, my main point that “if we cannot observe [samples] replicating or evolving: is this because they actually are incapable of growing or evolving, or because the conditions for observation are inappropriate?” remains valid, nonetheless.

- I am afraid that you affirmation that red blood cells are “alive” would meet with many very pertinent counter-arguments if exposed to the origin-of-life community ;-)

AUTHOR: This is an interesting topic, and I’ve expanded this section. The new paragraph reads as follows:

“Red blood cells (erythrocytes) require an active metabolism in order to maintain the integrity and function of the cell membrane and of the hemoglobin that holds the oxygen they transport. The cytoskeleton (with its associated ratcheting motor enzymes) is an essential component working to stabilize the membrane, but also maintain the correct flexibility. In the context of the above definition of life, erythrocytes function and live in an instantaneous sense, and die when the internal network of molecular motors ceases to function. Mammal erythrocytes do not include a nucleus and organelles, lacking some cell functions such as protein synthesis and oxidative phosphorylation, thereby limiting their autonomy and ability to persist. This has several advantages for mammal erythrocytes, including the ability to efficiently change shape as they pass through capillaries and, lacking the machinery required for replication, the superpower of invulnerability to viral infection. Aside from mammals and a few amphibians the erythrocytes of most animal groups do exhibit a nucleus and organelles. Bird erythrocytes, for example, have working mitochondria and fish erythrocytes are known to perform protein synthesis, although they do not replicate autonomously and are produced in an organ equivalent to the kidney (the opisthonephros). While it is undoubtedly correct to refer to the precursor cells of erythrocytes (normoblasts) as alive, mature erythrocytes should perhaps be seen as senescent (i.e., alive but no longer capable of a full suite of synthesis and replicative functions, and thus persistence). The same reasoning could be applied to other non-replicating cell types such as neurons. For example, a nervous system is alive but neuron function precludes mitosis and cellular replication, so the nervous system is inherently senescent; replication of the entire organism is required to generate a fresh nervous system. Organisms that lack nervous systems, such as plants, do not have the limitations (or advantages) of neurons, and can grow indeterminately.”

Author Response

N/A

Reviewer 3 Report

This paper is much more of an opinion piece as it is a review. As such, I found it very thought-provoking, and also very well-written. I found no less than 5 earlier versions of this paper on ArXiv ; the author has evidently been fine tuning his thinking and "sales pitch" for quite a while, and it shows.

The "life mechanism" advocated by the authors is very much a specific instantiation of Schrödinger's entropy reduction argument. The key is molecular "ratcheting" and constrained deformation associated with macromolecular dynamics: in physical parlance, folding is as transition, over a potential barrier, from a local free energy minimum to another lower energy state, generating ordered motional patterns through constraints and thus locally reducing entropy, all made possible by energy input from an external thermal reservoir.

One may well disagree, as I do, with some statements or opinions expressed by the author, yet I believe there should be room made for such papers in the technical literature. I am therefore inclined to recommend publication in Life.

I do list in what follows a number of suggestions/comments, as "food for further thought". These must be understood as coming from someone trained in physics and specializing in computational physics, astrophysical magnetohydrodynamics, and natural complex systems, including related origin of-life issues; but with very limited technical knowledge of biochemistry or macromolecular dynamics.

Specific comments:

The introduction section makes some rather unfair (in my opinion) statements regarding the current status of theories of life (e.g., lines 60-64, esp. "Theories of life do not dig beneath the surface"). Here a good overview of current theories would be: E. Trifonov (J. Biomolecular Struct. Dyn. 29, 259; 2011). Also of interest with regards to "digging beneath the surface": B. Daniels (Phys. Rev. Lett. 121, 138102; 2018), S. Walker & P. Davies (Roy. Soc. Interface 10, 20120869; 2013), and various recent research on life's energetics (more on this specific point further below). This is of course a huge topic, also inextricably entangled with origin-of-life issues, but I think the introduction should still provide a more balanced account.

Line 67: I suspect the authors meant to write "...but there are either *not* falsifiable..."

Section 5, lines 391-412: The discussion of ATP as a player in the author's proposed life mechanism has strong potential connections to current work of the energetics of life, e.g. the work by N. Lane and collaborators (e.g., in Cell 151, 1406, 2012; or in Bioessays 32, 271, 2010). Indeed the sentence at the end of the abstract: "Death is loss of integrated heat engine function" rang a strong bell in my mind, which I managed to trace back to the following sentence in the penultimate paragraph of N. Lane's (for the very educated) layman book: "Death is the ceasing of electron and proton flux..." (The Vital Question, Profile books, 2015). It seems to me that the author is missing here a great opportunity to connect his ideas to current research. I think it would be worth the effort.

Section 6: Two definitions of life are offered here (in italics and separated from the main body of the text). The first is far more specific than the second. The second presumably encompasses the first, but the connection is blurred by the fact that  the second definition also brings in two additional key components: self-replication and self-regulation (which the authors lumps under the term "autonomous"); and which, by extension, raises the question of origin. Yet in the paragraph following the second definition the authors seems to revert categorically to the "operation of heat engines" as "the single distinctive property distinguishing 'living' from 'non-living'...". What I am missing here ?

Section 7: I would take exception to some statements made here in the context of discussing specific "counterexamples" to the author's life definition. Cyclones are ultimately powered by a heat bath, namely the elevated oceanic surface temperatures driving thermal convection, with the Coriolis force setting the global patterning. Also, the bewilderingly complex patterns of snowflake growth takes place via a form of surface diffusion driven by the thermal energy from the surrounding air, a form of heat bath, and constrained by surface kinetics and attachment (see, e.g., K. Libbrecht, Rep. Prog. Phys. 68, 855-895, 2005; and references therein). I am not arguing that hurricanes or snowflakes are alive, but I believe they offer thornier counterexamples to the author's definition of life than the text actually suggests.

Section 8, lines 532-535: so this would be an instance of a system that would already be considered "alive" under the first definition of section 6, but would be deemed "alive" under the second definition of section 6 only *after* having "subsequently become self-regulating and self-replicating" ? This actually exemplifies the difficulty I had with section 6, as expressed above.

To sum up: while encouraging the authors to reflect upon the above comments (and perhaps adjust his paper accordingly), I am still giving a positive recommendation to this paper for publication in Life. "Big picture" papers are few and far in between, and this is one. It certainly got me thinking, and I would presume it may have the same effect on other readers; even should it proves in the end to have been off the mark.

Author Response

Reviewer 3

This paper is much more of an opinion piece as it is a review. As such, I found it very thought-provoking, and also very well-written. I found no less than 5 earlier versions of this paper on ArXiv ; the author has evidently been fine tuning his thinking and "sales pitch" for quite a while, and it shows.

The "life mechanism" advocated by the authors is very much a specific instantiation of Schrödinger's entropy reduction argument. The key is molecular "ratcheting" and constrained deformation associated with macromolecular dynamics: in physical parlance, folding is as transition, over a potential barrier, from a local free energy minimum to another lower energy state, generating ordered motional patterns through constraints and thus locally reducing entropy, all made possible by energy input from an external thermal reservoir.

One may well disagree, as I do, with some statements or opinions expressed by the author, yet I believe there should be room made for such papers in the technical literature. I am therefore inclined to recommend publication in Life.

I do list in what follows a number of suggestions/comments, as "food for further thought". These must be understood as coming from someone trained in physics and specializing in computational physics, astrophysical magnetohydrodynamics, and natural complex systems, including related origin of-life issues; but with very limited technical knowledge of biochemistry or macromolecular dynamics.

Specific comments:

The introduction section makes some rather unfair (in my opinion) statements regarding the current status of theories of life (e.g., lines 60-64, esp. "Theories of life do not dig beneath the surface").

AUTHOR: I have eliminated this – it was somewhat inflammatory and not conducive to future collaboration.

Here a good overview of current theories would be: E. Trifonov (J. Biomolecular Struct. Dyn. 29, 259; 2011). Also of interest with regards to "digging beneath the surface": B. Daniels (Phys. Rev. Lett. 121, 138102; 2018), S. Walker & P. Davies (Roy. Soc. Interface 10, 20120869; 2013), and various recent research on life's energetics (more on this specific point further below). This is of course a huge topic, also inextricably entangled with origin-of-life issues, but I think the introduction should still provide a more balanced account.

AUTHOR: The first article suggested by the Reviewer attempts to find truth from the frequency with which words are used in the ‘life’ literature. To simplify: if 560 out of 600 articles mention ‘growth’ then life must be about growth. By extension, if 93% of articles about cars mention acceleration, do we conclude that the underlying mechanism making cars go is acceleration? Or should we actually look at the internal combustion engine, drive chain and wheels (and the expansion of petrol when it is ignited and explosively changes phase)? Unfortunately, when I said that many papers only scratch the surface this is precisely what I meant. Living things are physical objects, and an explanation of those physical objects must lie at the fundamental level of their physical structure and functioning, not their emergent properties. The second suggested article underlines the importance of information for life, but again information must be physically encoded. I had already cited that Schrödinger (1944) noted the importance of repeated physical sequences for encoding information (i.e., that encoded information is fundamental to life) and I essentially stated that this wasn’t quite enough because it lacked a mechanistic explanation. Later on I also talked about information having a basis in the physical states of matter, so again information is important (and this was stated) but is underpinned by a physical mechanism.

In the revision I have toned down my criticism of some of the literature, but the entire article is essentially an attempt NOT to try and include every idea about life in general (much of which really is superficial), but specifically to review knowledge of the underlying physical mechanism, and make suggestions based on this. I felt that the Introduction had to deal with certain philosophical aspects of ‘life definition problems’ because much weight is often given to these and unfortunately purely philosophical arguments can stall mechanistic attempts and cut them off at the knee. I find this frustrating because much of modern science is predicated on the idea that the natural world has a basis in underlying physical processes, and that natural processes can be investigated. In contrast, continually invoking ‘life is something special and undefinable’ smacks of mysticism (to be clear, I am not criticizing the esteemed Reviewer!). In the revised manuscript, I have toned down this criticism, and I would like to keep a focus on reviewing the physical mechanism.

Line 67: I suspect the authors meant to write "...but there are either *not* falsifiable..."

AUTHOR: Indeed, this has been corrected.

Section 5, lines 391-412: The discussion of ATP as a player in the author's proposed life mechanism has strong potential connections to current work of the energetics of life, e.g. the work by N. Lane and collaborators (e.g., in Cell 151, 1406, 2012; or in Bioessays 32, 271, 2010). Indeed the sentence at the end of the abstract: "Death is loss of integrated heat engine function" rang a strong bell in my mind, which I managed to trace back to the following sentence in the penultimate paragraph of N. Lane's (for the very educated) layman book: "Death is the ceasing of electron and proton flux..." (The Vital Question, Profile books, 2015). It seems to me that the author is missing here a great opportunity to connect his ideas to current research. I think it would be worth the effort.

AUTHOR: I have now cited the two papers, but I’m afraid that in the few days I have available in which to conceive and write these revisions I will not have time to buy, read and incorporate the book. I agree that it would certainly have been useful and I’m a little ashamed that I don’t already have this on my bookshelves.

Section 6: Two definitions of life are offered here (in italics and separated from the main body of the text). The first is far more specific than the second. The second presumably encompasses the first, but the connection is blurred by the fact that  the second definition also brings in two additional key components: self-replication and self-regulation (which the authors lumps under the term "autonomous"); and which, by extension, raises the question of origin. Yet in the paragraph following the second definition the authors seems to revert categorically to the "operation of heat engines" as "the single distinctive property distinguishing 'living' from 'non-living'...". What I am missing here ?

AUTHOR: I think that some confusion has also been created by presenting two related definitions (for ‘life’ and ‘living thing’). Clearly the main objective is to elucidate the definition of life, and the second definition is a secondary concern, although a necessary one. Thus, I have decided not to put emphasis on the second definition and have incorporated it more subtly into the main text. For the record, the definition of life did also include the concept of self-regulation (“Life is a self-regulating process”), so this was not unique to the second definition (of a living thing). That said, the reviewer is correct that the second definition explicitly refers to heat engines, whereas the definition of life does not. The important point is not heat engines per se, but exploiting thermodynamic gradients (either directly or indirectly) using uniplanar conformation state changes. I have thus slightly altered the second definition to better ground it within the definition of life. Thus, a living thing is: “an autonomous network of units exploiting thermodynamic gradients to drive uniplanar conformation state changes that perform work”.

Section 7: I would take exception to some statements made here in the context of discussing specific "counterexamples" to the author's life definition. Cyclones are ultimately powered by a heat bath, namely the elevated oceanic surface temperatures driving thermal convection, with the Coriolis force setting the global patterning. Also, the bewilderingly complex patterns of snowflake growth takes place via a form of surface diffusion driven by the thermal energy from the surrounding air, a form of heat bath, and constrained by surface kinetics and attachment (see, e.g., K. Libbrecht, Rep. Prog. Phys. 68, 855-895, 2005; and references therein). I am not arguing that hurricanes or snowflakes are alive, but I believe they offer thornier counterexamples to the author's definition of life than the text actually suggests.

AUTHOR: See the new paragraph on the thermodynamics of classical heat engines, in which I now explicitly present these examples (and later show why they do not conform to the definition of life). I have used the suggested citation.

Section 8, lines 532-535: so this would be an instance of a system that would already be considered "alive" under the first definition of section 6, but would be deemed "alive" under the second definition of section 6 only *after* having "subsequently become self-regulating and self-replicating" ? This actually exemplifies the difficulty I had with section 6, as expressed above.

AUTHOR: The concept of ‘self-regulating’ was present in both definitions (see discussion above) so these are not contradictory.

To sum up: while encouraging the authors to reflect upon the above comments (and perhaps adjust his paper accordingly), I am still giving a positive recommendation to this paper for publication in Life. "Big picture" papers are few and far in between, and this is one. It certainly got me thinking, and I would presume it may have the same effect on other readers; even should it proves in the end to have been off the mark.

AUTHOR: Thank you for keeping an open mind!

Reviewer 4 Report

I am thrilled to accept your manuscript, but please consider including works by or about Alfonso Luis Herrera as he is a crucial figure in the discussions of the definitions of life in the early 20th century. I would like to include references to some of his works in the corrected manuscript.

Alfonso Luis Herrera and the Beginnings of Evolutionism and Studies in the Origin of Life in Mexico J Mol Evol

. 2016 Dec;83(5-6):193-203. doi: 10.1007/s00239-016-9771-7 

Cleaves II HJ, Lazcano A, Ledesma Mateos I, Negrón-Mendoza A, Peretó J, Silva E (2014). Herrera's 'Plasmogenia' and Other Collected Works: Early Writings on the Experimental Study of the Origin of Life. Springer. ISBN 978-1493907366.

Negrón-Mendoza, A. (1995). "Alfonso L. Herrera: A Mexican pioneer in the study of chemical evolution". Journal of Biological Physics. 20 (1–4): 11–15. doi:10.1007/BF00700417

Herrera, A.L. (1942): A New Theory of the Origin and Nature of Life,Science,96, 2479.

Author Response

Reviewer 4:

I am thrilled to accept your manuscript, but please consider including works by or about Alfonso Luis Herrera as he is a crucial figure in the discussions of the definitions of life in the early 20th century. I would like to include references to some of his works in the corrected manuscript. 

 Alfonso Luis Herrera and the Beginnings of Evolutionism and Studies in the Origin of Life in Mexico J Mol Evol. 2016 Dec;83(5-6):193-203. doi: 10.1007/s00239-016-9771-7 

 Cleaves II HJ, Lazcano A, Ledesma Mateos I, Negrón-Mendoza A, Peretó J, Silva E (2014). Herrera's 'Plasmogenia' and Other Collected Works: Early Writings on the Experimental Study of the Origin of Life. Springer. ISBN 978-1493907366.

 Negrón-Mendoza, A. (1995). "Alfonso L. Herrera: A Mexican pioneer in the study of chemical evolution". Journal of Biological Physics. 20 (1–4): 11–15. doi:10.1007/BF00700417

 Herrera, A.L. (1942): A New Theory of the Origin and Nature of Life,Science,96, 2479.

AUTHOR: This is an excellent suggestion, not just from an historical point of view and because these ideas should be cited, but also because the main idea of Herrera is that life represents instantaneous physicochemical processes occurring in the protoplast, which (although lacking the detailed modern view of the functioning of components within the protoplast) is in clear agreement with the more detailed definition of life proposed in the current manuscript. Unfortunately I did not have time or space enough to discuss this in extensive detail, but I have now cited Herrera’s original book (Herrera 1932) where he laid out his ideas, and have included a few relevant quotes. The reviewer’s suggestion has certainly improved the manuscript.

Round 2

Reviewer 1 Report

The author has removed most of the parts which I had some difficulties with in the original manuscript. He didn't accept my opinion that ribozymes are irrelevant to the present discussion. While I do not intend to force the author into accepting my point of view, I would nevertheless communicate to the author my opinion on the matter. The review from Marie-Christine Maurel's group which you cite to substantiate your point just as all other papers I have read on this topic only strengthen my opinion that the ribozyme story is  bonkers. All these ribozymes do only one thing and that is operate on themselves performing phosphoryl-transfer reactions. That is clearly not what you need to make a metabolism! The often hailed sole example of a different reaction is in the ribosome. Unfortunately, all these proponents of an RNA-world forget to mention that what is crucial in the heart of the ribosome where the chain-elongation occurs is a peptide!!! I therefore hold my case that there is not a single ribozyme in biology that does any catalysis meaningful to metabolism (and that is what you are talking about when you make your thermodynamic statements). Still, as I said, I don't object your perpetuating the decades-old "views" of the RNA-world people, that's what many papers do ...

As I noted in my first review, I am ignorant on the other examples the author brought forward. The point that those that I know something about didn't really fit the author's claim (and he has accordingly now removed them), makes me admittedly a bit worried about the actual pertinence of the others. Nevertheless, I will opt for the benefit of doubt and advocate publication of this ms. However, given the remaining doubts, I would suggest that the article type be Hypothesis or Opinion, rather than Review.

A final comment to the author: Your new discussion of ATP synthase now discusses the directionality of motion of protons determined by a concentration gradient. In real-life situations this actually isn't the case. Both sides of the energy-converting membrane are so heavily buffered that there is no measurable proton gradient (apart from in thylakoids but their small deltapH  is a secondary effect from potassium-efflux systems). However, there is an electrostatic field across this membrane, which will pull protons through and even generate a proton-surplus on the negative-side. This deltaPsi of course is also a "thermodynamic gradient", while the author's description seems to see thermodynamic gradients only in Brownian terms.

Author Response

The author has removed most of the parts which I had some difficulties with in the original manuscript. He didn't accept my opinion that ribozymes are irrelevant to the present discussion. While I do not intend to force the author into accepting my point of view, I would nevertheless communicate to the author my opinion on the matter. The review from Marie-Christine Maurel's group which you cite to substantiate your point just as all other papers I have read on this topic only strengthen my opinion that the ribozyme story is  bonkers. All these ribozymes do only one thing and that is operate on themselves performing phosphoryl-transfer reactions. That is clearly not what you need to make a metabolism! The often hailed sole example of a different reaction is in the ribosome. Unfortunately, all these proponents of an RNA-world forget to mention that what is crucial in the heart of the ribosome where the chain-elongation occurs is a peptide!!! I therefore hold my case that there is not a single ribozyme in biology that does any catalysis meaningful to metabolism (and that is what you are talking about when you make your thermodynamic statements). Still, as I said, I don't object your perpetuating the decades-old "views" of the RNA-world people, that's what many papers do ...

AUTHOR: Perhaps there is not a single ribozyme in biology today that does any catalysis meaningful to the metabolism of modern cells, but the metabolism of extant cells is likely to be very different from the organisms which occurred before the Last Universal Common Ancestor, and which relied on different systems with respect to surviving lineages. One of the main conditions for the divergence of life into the three main domains was the evolution of efficient ribosomes (https://www.ncbi.nlm.nih.gov/pmc/articles/PMC430263/ ) so the life that came immediately before the LUCA probably had RNA-based mechanisms that were simpler than ribosomes as we know them today (perhaps they did not involve peptide linkage, but had a different function). It is not surprising that purely RNA-based metabolism is not evident in modern organisms, because it was superseded, out-competed, and lost by the time of the LUCA and thus will not be evident in the metabolisms of the three extant domains of organisms. But ribozymes are molecules capable of conformation state changes involved in activity, they function in a way that agrees with other examples of biomolecular ‘machines’, and are at least consistent with having had biological roles. Also, modern ribosomes do exhibit uniplanar conformation changes which induce linear motion in a system that performs work, and agree with the definition of life suggested here. I have now added the following text to the revised manuscript (line 273) which points out the difference in roles of ribozymes between extant and ancient organisms and is more cautious concerning their function in modern cells:

“While catalytic activity is mediated by enzymes in extant organisms, the universal presence of ribozymes across the domains of life suggests that they may have been crucial to catalysis for the organisms that preceded the Last Universal Common Ancestor (LUCA) of extant life [41]”.

As I noted in my first review, I am ignorant on the other examples the author brought forward. The point that those that I know something about didn't really fit the author's claim (and he has accordingly now removed them), makes me admittedly a bit worried about the actual pertinence of the others. Nevertheless, I will opt for the benefit of doubt and advocate publication of this ms. However, given the remaining doubts, I would suggest that the article type be Hypothesis or Opinion, rather than Review.

  • AUTHOR: The reviewer states that I have removed ‘them’ (plural), referring to examples of conformation state changes in biomolecules. I only removed the single example of chlorophyll. The ribozymes are, as acknowledged above, still included, as are the rotary enzymes, which have a more complete description of their structure and discussion of their thermodynamics, and which remain pertinent to the definition.

A final comment to the author: Your new discussion of ATP synthase now discusses the directionality of motion of protons determined by a concentration gradient. In real-life situations this actually isn't the case. Both sides of the energy-converting membrane are so heavily buffered that there is no measurable proton gradient (apart from in thylakoids but their small deltapH  is a secondary effect from potassium-efflux systems). However, there is an electrostatic field across this membrane, which will pull protons through and even generate a proton-surplus on the negative-side. This deltaPsi of course is also a "thermodynamic gradient", while the author's description seems to see thermodynamic gradients only in Brownian terms.

AUTHOR: I had originally concentrated on the chemical (proton) gradient, because of the obvious link with diffusion and thus the kind of thermodynamic processes that are generally invoked throughout the manuscript. My aim was to underline that ATP synthase exploits thermodynamic gradients. However, yes, both the chemical gradient and the electric gradient are components of the proton-motive force, and I should have also explicitly named this (which I now have in the revision). The revised text also includes the name of the process driven by the proton-motive force, namely chemiosmosis, to be complete and correct.

Reviewer 2 Report

I've looked at this version. I see no real change from my original comments

Author Response

Report

Concerning Dr. Pierce's work entitled Life's Mechanism and submitted to Life Journal.

As far as I am concerned, this work is an attempt to promote the idea of a self-regulating system with cycles and out of equilibrium within the framework of so-called complex systems (Autopoiesis, H Maturana). In the understanding that it is a condition of what could be thought by "life". An interesting work of compilations tends to unify the problem.

It is a vision that emphasizes the idea of a "heat engine" and therefore thermodynamic aspects out equilibrium (it is interesting).

The serious problem is that it is not quantitative as thermodynamics is. Therefore, some assertions are difficult to refute or admit.

For example, in section 6, life is defined (421) as "life is ...  … locally reduces Entropy" (422-425). My refrigerator meets that criterion, and I don't think it's alive (and I hope so).

The second law says that in an isolated system, the variation of entropy never decreases. Beware the entropy is not negative, which is a systematic error of the author (negative entropy).

Back to the point. Let A be the local system (alive), B the environment, the second principle:

In the case of the most orderly system ("being alive")  and the above inequality must be satisfied (second principle). A refrigerator satisfies that criterion, in fact,   and . As long as  the heat is supplied to the environment  (The environment gets disordered). That is, according to the first principle, work (energy) must be delivered to the system. In the framework of the manuscript, this is the input ("related with aggregate matters to produce negative entropy")

In short, it is what is expected: quantitative aspects in a correct formulation of "Life's Mechanism".

Finally, as mentioned, the rigorous definition of entropy (Boltzmann) is never negative (as the author maintains, for example in Rule 416 and others). This is  with N number of possible states in the system.

AUTHOR: Firstly, my apologies for not having responded during the first round of review, but I did not see the word document and it had been indicated to me that this particular review was ‘spurious’, which I understood to represent an error in the system. During the second round I saw the comment that this review had not been taken into account, so I asked to be given a chance to respond.

With regard to entropy, I presented Erwin Schrödinger’s (1944) work and his terminology (“negative entropy” or “negentropy”). I did not use the term “negative entropy” in the definition of life, and it is incorrect that “the author maintains” that life involves negative values of Boltzmann’s entropy (i.e. breaks the second law of thermodynamics). The definition of life I suggested states that life “locally reduces entropy” (i.e. positive values become less positive) and this was discussed as the sum of negative and positive entropy processes. Notably, I cited literature [54, 55, also 25] to specifically state that the second law of thermodynamics is not contravened: “Despite reducing entropy locally, heat engines do not contravene the second law of thermodynamics (that entropy in a system always increases), because the work they perform represents a relatively small decrease in entropy connected to and driven by a larger entropy increase: i.e., a localized decrease but a net increase.”.

Essentially, the equation presented by the reviewer is in agreement with this but actual calculations would require further internal terms for various aspects of the system that each make up SA and SB. These internal terms are far too complex to actually provide. To illustrate this, I can use the example of a highly simplified model plant and the resources it uses for growth: say there are a large number of molecules of CO2 dispersed throughout a liter of air, which enter a leaf, are assimilated by photosynthesis to form carbohydrate molecules, which are then used to produce cellulose microfibrils and cell wall material. The CO2 molecules were dispersed throughout a volume of air (a relatively disordered state), but are now bonded together to form a compact, solid structure (a relatively ordered state). For the CO2 molecules, and the plant, this is a local increase in order (local entropy reduction) and represents one entropy term. However, this is not the entire system. In the atmosphere/sun/soil/plant system the resources light, CO2 and water are not just converted into cells, but also into waste products including O2, protons and heat. There is an overall increase in entropy because a range of processes involve increasing disorder (the production of photons by fusion, the fluxes and diffusion of water in the soil and atmosphere, the fluxes and diffusion of CO2 in the air, the diffusion of protons from root to soil, re-emission and radiation of photons not used in photosynthesis, and then the dissipation of heat by transpiration) despite the small local ordering of matter evident as the plant body. In other words, the growing plant exhibits locally reduced entropy in the context of an overall environmental increase. This is essentially the system that the reviewer wishes me to summarize with a simple equation. I suggest that this is not necessary because: 1). local entropy reduction, as per the definition of life, is undoubtedly real because organisms do aggregate sparse resources into solid bodies (increasing order does occur in part of the system and we can literally see this as the bodies of the organisms themselves - to deny this is to deny that organisms exist as corporeal objects), 2). the idea being presented here, and underpinning the definition of life, is not concerned with proving entropy reduction but is actually more concerned with how biological molecules respond to thermal energy and move to perform work (i.e. the mechanism by which they operate). This is not addressed by the reviewer.

Finally, a refrigerator, in the context of the suggested definition, would certainly not be considered alive because the definition is not based solely on entropy, and considers the autonomous network of heat engines (not the single heat engines) as alive. I had already dealt with this in the following text which uses the example not of a refrigerator but of a loom: “Autonomy and self-regulation via integrated networks are key concepts highlighted in this definition. Looms use cyclic conformation changes (mechanical action) to convert energy and matter (electricity and wool) into an ordered state (cloth) following a pattern encoded as a set of instructions (programmed information). However, looms are not self-regulating systems and require external input (from a biological organism) for their creation, maintenance, operation and programming. In other words, it is not the single protein or ribozyme (the single heat engine) that should be considered alive, but the integrated, self-regulating and self-replicating network of heat engines.”

Round 3

Reviewer 2 Report

The answers are not convincing in relation to my report. I stand by my original decision.

Author Response

The answers are not convincing in relation to my report. I stand by my original decision.

AUTHOR: The reviewer expects quantification (from their previous review: "what is expected: quantitative aspects in a correct formulation of Life's Mechanism"), but they do not address the main definition itself which is based on the concept that key biological molecules change conformation in a way that directs thermal disequilibria into work. This is concerned with the mechanism, not how to quantify a value for “life”. In other words, it is like me stating “a car works on the principle of internal combustion, which ultimately drives wheels to move the car” but the reviewer replying “You can’t say that unless you show a mathematical formula”. Nonetheless, it is useful to know about internal combustion and how it can be directed to make the wheels turn, and it would be a shame if car mechanics could never write about cars because they don’t describe them mathematically. It is also difficult to understand how the mechanistic statement “Life is a self-regulating process whereby matter undergoes cyclic, uniplanar conformation state changes that convert thermodynamic disequilibria into directed motion, performing work that locally reduces entropy” can be reduced to any kind of single value that would have any meaning. Aside from ignoring the actual mechanistic process being discussed, the reviewer’s entire premise is that the concept of negative entropy is wrong, but this is not my concept, it is Erwin Schrödinger's. The onus is on the reviewer to disprove Erwin Schrödinger.